# Effect of Homo-Fermentative Lactic Acid Bacteria Inoculants on Fermentation Characteristics and Bacterial and Fungal Communities in Alfalfa Silage

**Yanbing Li [1,2,3], E. B. da Silva [4], Jingchun Li [1,2,3] and L. Kung, Jr. [4,*]**

1. College of Animal Science and Veterinary Medicine, Heilongjiang Bayi Agricultural University, Daqing 163319, China
2. Key Laboratory of Low-Carbon Green Agriculture in Northeastern China, Ministry of Agriculture and Rural Affairs, Heilongjiang Bayi Agricultural University, Daqing 163319, China
3. Key Laboratory of Efficient Utilization of Feed Resources and Nutrition Manipulation in Cold Region of Heilongjiang Province, Daqing 163319, China
4. Department of Animal and Food Sciences, University of Delaware, Newark, DE 19716, USA
* Correspondence: lksilage@udel.edu; Tel.: +1-30-2388-5589

**Abstract:** We evaluated the effects of a homo-fermentative lactic acid bacteria (homo-LAB) inoculant on the fermentation and microbial communities of alfalfa ensiled at two dry matter (DM) contents of 38 and 46% DM. At both DMs, alfalfa was treated or not with an inoculant containing *Pediococcus acidilactici*, *Enterococcus faecium* and *Lactobacillus plantarum* at a targeted application rate of 165,000 cfu/g of fresh weight and stored for 3, 30 and 60 days. Treatment with the inoculant resulted in a lower drop in pH and, in general, higher lactic acid and lower acetic acid when applied to medium DM silage. For the four most abundant microbial genera, increased abundances of *Bacteroides* and *Lactobacillus* ($p < 0.05$), as well as decreased abundances of *Muribaculaceae* were observed in high DM and inoculated silages. The abundance of *Prevotellaceae-UCG-001* was lower in medium DM control silages than in high DM control silages. Inoculation and DM affected abundances of *Vishniacozyma* ($p < 0.05$). Increased abundances of *Vishniacozyma*, as well as decreased abundances of *Leucosporidium* were observed in medium DM-inoculated silages. Changes in the relative abundance (RA) of the main populations of bacteria and yeasts did explain the fermentation and nutrition differences among treatments.

**Keywords:** microbial inoculants; alfalfa silage; dry matter; microbial community; relative abundance

## 1. Introduction

Alfalfa is one of the popular fodder crops fed to ruminants, but high-quality alfalfa silage is difficult to make due to its high buffering capacity, low concentration of soluble carbohydrates and the presence of epiphytic lactic acid bacteria in the raw material [1]. Epiphytic and endophytic harmful bacterial and fungi constitute the predominant microbiota as green fodder, and a substrate converted to silage is highly susceptible to decrease silage quality, negatively affecting the health status of animals [2]. Silage inoculants containing lactic acid bacteria (LAB) are not always successful at improving silage quality because the degree of wilting, ensiling times and environmental temperatures vary greatly and can restrict the success of these inoculants [3,4]. Better-adapted epiphytic microflora might outcompete organisms for inoculant supply and dominate the succeeding fermentation [5]. Increasing the dry matter content of forage at ensiling can affect the ensuing fermentation, proteolysis and respiration in the silo. For example, in alfalfa silages wetter than 40% DM, Stallings et al. (1981) and Papadopoulos and McKersie (1983) reported increased proteolysis than in more highly wilted silages, while, in drier silages (greater than 40% DM), increased wilting has consistently reduced proteolysis [6,7].

Silages treated with homo-fermentative lactic acid bacteria (homo-LAB) have been shown to improve fermentation quality, indicated by a fast lowering of pH, high lactic acid content and low level of ammonia-N [8–11]. These marked features of a silage inoculant arise from the proliferation of homo-LAB at the expense of hetero-LAB and other species [12]. The use of homo-LAB such as *L. plantarum*, *E. faecium* and *P. acidilactici* is common [13]. *L. plantarum* is widely used as a silage additive because the species effectively utilizes WSCs to produce lactic acid, which rapidly reduces the pH of the ensiled mass [14]. Li reported that ferulic acid could be released from plant cell walls after treatment of *L. plantarum* A1; the release of ferulic acid from plant cell walls may increase the digestion of whole corn silage in the rumen [15].

Additives for ensilage, consisting of several homo-LAB strains, may have a different impact on the ensilage process; as we know, it is important to study the use of different combinations of LABs to determine the most beneficial effect on the different types of silages. More studies are needed to evaluate the effectiveness of homo-LAB combinations on affecting the fermentation of the kinds of silages under challenging management issues that can occur in the field, such as dry matter, storage times and any interactions among these factors.

Next-generation sequencing (NGS) technology has become more sophisticated and easily accessible to characterize the microbial communities associated with silage fermentation [16]. Characterizing the changes of the microbial community and its ultimate effects on inoculant interactions with pre-wilting might provide valuable information for improving silage fermentation with homo-LAB inoculation [8]. The fact that most assessment of inoculants have focused only on fermentation products may lack insights into the bacterial and fungal microbiota associated with homo-LAB and any interactions with dry matter and storage times. The efficacy of homo-LAB inoculant use should be evaluated, not only for fermentation products, but also for the effects on microbial communities. The aim of this study was to evaluate the effect of three *Lactobacillus* species on the chemical and microbiological parameters of alfalfa silage by evaluating the effect of applying a homo-LAB inoculant on the fermentation and microbial community of alfalfa silage harvested at two DM contents.

## 2. Materials and Methods

### 2.1. Ensiling

On 26 June 2017, second-cut alfalfa (early bloom stage) from one field was harvested at the University of Delaware farm in Newark, Delaware and wilted to a DM content of 37.8% or 46.5%. Forage was chopped to 0.95 cm theoretical chop and five individually replicated piles of forage at each DM were treated with: a) no treatment (CTR) or b) treatment with Purina FI Enhance Inoculant (PEI, Purina Animal Nutrition LLC, Arden Hills, MN, USA) containing *Pediococcus acidilactici*, *Enterococcus faecium* and *Lactobacillus plantarum*, with a final targeted application rate of 165,000 cfu of total LAB/g of fresh forage weight. After treatment, about 1 kg of forage was packed into nylon–polyethylene vacuum barrier bags (3.5 mil thickness, $15.2 \times 30.5$ cm$^2$, O$_2$ transmission rate of 97 cm$^2$/m$^3$/d at 22.8 °C, 0% rh; Doug Care Equipment Inc., Springville, CA, USA) for each treatment. Air was vacuum-removed from the bags and heat-sealed using a Best Vac vacuum sealer (distributed by Doug Care Equipment Inc., Springville, CA, USA). All silos were stored at $21 \pm 0.5$ °C for 3, 30 and 60 d.

### 2.2. Analytical Procedures

We measured the DM content and pH of fresh forages from four CTR piles at 0 d and of five samples from each silo after ensiling. The DM content was determined after 48 h of incubation in a 60 °C forced air oven. Weights of full and empty silos were measured at silo opening and with the DM content of fresh and ensiled samples, which were used to calculate DM recovery. Representative 25 g portions of fresh forage and silage were mixed with 225 mL of sterile, quarter-strength Ringers solution (Oxoid BR0052G, Oxoid,

Unipath Ltd., Basingstoke, UK) and homogenized for 1 min in a Proctor-Silex 57171 blender (Hamilton Beach/Proctor-Silex Inc., Washington, DC, USA). The homogenate was filtered through four layers of cheesecloth. The pH was determined on the homogenized samples using an Oakton pH700 Meter (Oakton Instruments, Vernon Hills, IL, USA).

Portions of the water extracts previously filtered through four layers of cheesecloth were further filtered through Whatman 54 filter paper (Whatman Inc., Clifton, NJ, USA). The extract was acidified with $H_2SO_4$ 50% vol/vol and frozen. Later, the extract was analyzed for the concentrations of lactic, acetic, butyric and propionic acids and 1, 2- propanediol and ethanol [17] by high-performance liquid chromatography using a Shimadzu LC-20AD (Shimadzu Scientific Instruments, Columbia, MD, USA). The concentration of NH3-N was determined in the water extracts by the phenol–hypochlorite method of Weatherburn (1967) [18], and water-soluble carbohydrates (WSC) were determined by the colorimetric procedure of Nelson (1944) [19].

Nutrient analysis of fresh forage described below was performed by Cumberland Valley Analytical Services (Waynesboro, WV, USA). A portion of each dried sample was ground using an Udy Cyclone Sample Mill (Udy Corp., Fort Collins, CO, USA) to pass through a 1 mm screen. Total N was measured by combustion of the sample following the AOAC method 990.03 (AOAC, 2010) [20] using a Leco CNS2000 Analyzer (Leco Corporation, St. Joseph, MI, USA), and the concentration of crude protein (CP) was calculated by multiplying the resulting total N concentration by 6.25. Soluble protein (SP) was determined by the method of Krishnamoorthy et al. (1982) [21] and expressed as % of CP. The concentration of acid detergent fiber (ADF) was quantified on dried ground samples using the AOAC method 973.18 [20], with the modification of using Whatman 934-AH glass microfiber filters with 1.5 μm particle retention instead of fritted glass crucibles. The concentration of neutral detergent fiber (NDF) was analyzed on dried samples according to the methodology of Van Soest et al. (1991) [22], with sodium sulfite and amylase and was not corrected for ash content. Another portion of the dry sample was ground to pass a 3 mm screen and analyzed for starch by the methodology of Hall (2008) [23].

### 2.3. Enumeration of Microorganisms on Agar Plates

For each sample, a portion of the previously described, freshly prepared water extract was serially diluted in quarter-strength Ringers solution, and the numbers of LAB were determined by pour-plating in de Man, Rogosa and Sharpe (MRS) agar (CM3651, Oxoid, Unipath Ltd., Basingstoke, UK). Plates were incubated anaerobically at 35 °C in sealed plastic containers with anaerobic packs (Mitsubishi Gas Chemical Co., Tokyo, Japan) and an anaerobic indicator (BR0055, Oxoid, Unipath Ltd., Basingstoke, UK). Containers were vacuum-sealed in nylon–polyethylene bags (3.5 mil embossed pouches, $15.2 \times 30.5$ cm$^2$; Doug Care Equipment Inc., Springville, UT, USA) using a Best Vac vacuum machine (distributed by Doug Care Equipment Inc.). Plates with a number of colonies between 30 and 300 were counted after 48 and 72 h to obtain the number of colony-forming units. Numbers of yeasts and molds were determined on malt extract agar (MEA, CM0059, Oxoid, Unipath Ltd., Basingstoke, UK) acidified after autoclaving with lactic acid (85%) at a rate of 0.5% vol/vol. The plates were incubated aerobically at 30 °C and enumerated after 48 and 72 h. Plates with a number of colonies between 30 and 300 were counted to obtain the number of colony-forming units.

### 2.4. Analysis of the Composition of the Microbial Communities by Next-Generation Sequencing

Samples from five replicates of fresh forage and 90 d silages (CTR, PEI) were analyzed for the composition of their microbial community by NGS. Representative 25 g portions of the samples were mixed with 225 mL of autoclaved quarter-strength Ringers solution (Oxoid BR0052G, Oxoid, Unipath Ltd., Basingstoke, UK) for 2 min using a Colworth 400 stomacher (Seward, London, UK). The homogenates were filtered through four layers of sterile cheesecloth. Next, 2 mL of the filtrate were centrifuged for 3 min at $21,000 \times g$ using a Centrifuge 5424 R (Eppendorf AG, Hamburg, Germany). The supernatant was

discarded, and 100 μL of autoclaved Ringers solution was used to resuspend the pellet. Samples were kept at −80 °C for further analysis.

Extraction of DNA, amplification of the 16S rRNA and internal transcribed spacer 1 (ITS1) and Illumina MiSeq-based sequencing were performed by the Research and Testing Laboratory (Lubbock, TX, USA). DNA was extracted using the MoBio PowerMag Soil kit (MoBio Laboratories Inc.,Carlsbad, CA, USA) according to the manufacturer's instructions. For bacterial analysis, the primers 515F (5′-GTGCCAGCMGCCGCGGTAA-3′) and 926R (5′-CCGTCAATTCMTTTRAGTTT-3′) [24] were used to amplify the V4 and V5 hypervariable regions of the 16S rRNA gene. For fungal analysis, the primers ITS1F (5′-CTTGGTCATTTAGAGGAAGTAA-3′) and ITS2aR (5′-GCTGCGTTCTTCATCG ATGC3′) [25,26] were used to amplify the ITS1. Sequencing was performed on an Illumina MiSeq (San Diego, CA, USA) platform using the 2 × 250 bp paired-end method.

The procedure of *Next-Generation Sequencing* analysis was followed by E. B. da Silva (2020) [27]. All sequencing data were received as FASTQ files and deposited in the NCBI Sequence Read Archive under BioProject accession PRJNA886387. We have released the above data.

### 2.5. Statistical Analysis

Agar counts of lactic acid bacteria and yeasts and molds were transformed to log10 before statistical analysis and were presented on a fresh weight basis. Data were analyzed using JMP version 12 (SAS Institute, Cary, NC, USA). Data on the chemical, microbial and physical parameters were analyzed as a 3 × 2 × 2 factorial arrangement of treatments in a completely randomized design. The model included the following main effects: day of ensiling (DAY), effect of inoculation (INO) and effect of DM (DRM). The interactions of DAY × INO, DAY × DRM, INO × DRM and DAY × INO × DRM were tested. The number of statistical repetitions for each group was 5. Differences were considered significant when $p < 0.05$ and trends towards significance were considered when $p < 0.10$. Pairwise mean comparisons were performed using Tukey's test at alpha = 0.05. [28]. For microbial composition analysis, at each taxonomic level, we defined the taxa with relative abundance > 0.01% in at least one sample as identified, while those with relative abundance > 0.1% presented in more than half of the sample per group as detected.

## 3. Results

### 3.1. Targeted and Actual LAB Application Rate

The targeted and actual final application rates of LAB added to the forage prior to ensiling are shown in Table 1. The amount of LAB applied was slightly lower than planned.

**Table 1.** Targeted and actual final lactic acid bacteria application rates.

| Treatment | Targeted Final Lactic Acid Bacterial Targeted Application Rate—cfu/g of Fresh Forage | Actual Final Lactic Acid Bacterial Targeted Application Rate—cfu/g of Fresh Forage |
|---|---|---|
| Medium dry matter alfalfa PEI [1] | 165,000 | 150,482 |
| High dry matter alfalfa PEI | 165,000 | 106,533 |

[1] PEI—inoculated with Purina FI Enhance Inoculant (PEI, Purina Animal Nutrition LLC, Arden Hills, MN, USA). Actual final lactic acid bacterial targeted application rate based on plating the inoculant solutions actually applied on the day of the study.

### 3.2. Nutrient Composition and Numbers of Culturable Microorganisms in Fresh Forage

The characteristics of the freshly chopped alfalfa before ensiling are presented in Table 2. Before ensiling, fresh and untreated alfalfa were wilted to a DM content of 378 g/kg (low DM) and 461 g/kg (high DM), and the pH value was 6.24 and 6.23. The average number of epiphytic microbes in fresh alfalfa such as LAB, yeasts and molds were about

5.86, 3.33 and 2.95 log10 cfu/g, respectively, and did not differ among the two DM level silage materials. Wilted alfalfa was low in water soluble carbohydrates and nitrates, high in protein and had a high buffer capacity at a fermentation coefficient (FC) below 58.

**Table 2.** Chemical (% DM basis unless stated otherwise) and microbial analysis (log10 cfu/g fresh weight basis) of freshly chopped alfalfa before treatment.

| Item | Medium DM Alfalfa | High DM Alfalfa | SEM * |
|---|---|---|---|
| Dry matter | 37.82 | 46.05 | 1.46 |
| pH | 6.24 | 6.23 | 0.01 |
| Crude protein | 17.80 | 17.60 | 0.13 |
| Soluble crude protein, % of crude protein | 45.07 | 40.11 | 2.10 |
| Ammonia nitrogen | 0.03 | 0.03 | <0.01 |
| Acid detergent fiber | 35.58 | 35.04 | 0.19 |
| Neutral detergent fiber | 43.38 | 43.46 | 0.23 |
| Ash | 9.05 | 9.07 | 0.09 |
| Water soluble carbohydrates | 8.06 | 8.07 | 0.11 |
| Buffering capacity (gm lactic/100 g dry matter) | 5.56 | 5.65 | 0.04 |
| Lactic acid bacteria | 5.83 | 5.92 | 0.07 |
| Yeasts | 3.06 | 3.62 | 0.20 |
| Molds | 3.29 | 2.67 | 0.13 |

* SEM—the standard error of mean difference, n = 5.

### 3.3. Nutrient Composition, Fermentation Profile and Numbers of Culturable Microorganisms in Silages

As expected, because fermentation end products change with the time of ensiling, there were numerous three-way interactions in this study. Results and discussion will focus primarily on the effects of inoculant treatments and any interactions between DM and days of ensiling.

The chemical composition of silages, shown as the three-way interaction, is shown in Table 3, and only appropriate significant main and two-way interactions are shown (when there was no overlapping three-way or two-way interactions) in Table 4. As expected, the concentration of DM was not affected by inoculation, but it was different between DM (36.96 vs. 45.20% DM) and, on average, was slightly lower at 60 d (40.47%) compared to 3 (41.43%) and 30 d (41.44%) (Table 4). There were only main effects for the concentration of CP in silage. The concentration of CP was greater for PEI (average of 18.16%) vs. CTR (17.94%), lower for high DM silages (17.92%) compared to medium DM silages (18.25%) and increased with days of ensiling (17.87% at 3 d, 18.07% at 30 d and 18.26% at 60 d) (Table 4). There was a three-way interaction for Sol-CP (Table 3). Inoculation had no effect compared to CTR at 3 and 30 d. There was also a three-way interaction for $NH_3$-N (Table 3). Similar to Sol-CP, inoculation had no effect when compared to CTR at 3 and 30 d. However, after 60 d in medium DM silage, inoculation resulted in less $NH_3$-N (0.211% for PEI) compared to CTR (0.245%). In high DM silage at 60 d, inoculation numerically reduced $NH_3$-N compared to CTR. INO and DRM affected ADF, but differences among treatments were not related to their interaction. The concentrations of NDF were higher in high DM than medium DM samples (Table 4). The concentration of WSC (Table 3) was higher in inoculated silages (PEI = 6.37%) than in CTR (4.44%) in the final 60 d in medium DM silage but was generally unaffected by inoculation at other times (three-way interaction).

**Table 3.** The DM and chemical analysis (% DM basis unless stated otherwise) of alfalfa silage ensiled for 3, 30 and 60 d. Means shown for three-way interaction. (ADF, NDF and ash data only for day 60).

| Treatments | Dry Matter | Crude Protein | Soluble Crude Protein, % of Crude Protein | Ammonia Nitrogen | Acid Detergent Fiber | Neutral Detergent Fiber | Ash | Water Soluble Carbohydrates |
|---|---|---|---|---|---|---|---|---|
| **Day 3** | | | | | | | | |
| **Medium dry matter alfalfa** | | | | | | | | |
| CTR * | 38.82 [b] | 17.76 [b] | 50.46 [d] | 0.11 [d] | nd | nd | nd | 6.85 [c] |
| PEI | 36.50 [b] | 18.24 [a] | 50.96 [d] | 0.12 [d] | nd | nd | nd | 7.01 [bc] |
| **High dry matter alfalfa** | | | | | | | | |
| CTR | 45.51 [a] | 17.74 [b] | 49.74 [d] | 0.11 [d] | nd | nd | nd | 8.31 [a] |
| PEI | 45.51 [a] | 17.74 [b] | 48.56 [d] | 0.11 [d] | nd | nd | nd | 8.11 [ab] |
| **Day 30** | | | | | | | | |
| **Medium DM alfalfa** | | | | | | | | |
| CTR | 37.06 [b] | 18.16 [ab] | 61.14 [bc] | 0.23 [ab] | nd | nd | nd | 3.50 [efg] |
| PEI | 36.54 [b] | 18.32 [a] | 59.18 [c] | 0.22 [c] | nd | nd | nd | 4.95 [d] |
| **High dry matter alfalfa** | | | | | | | | |
| CTR | 46.21 [a] | 17.74 [b] | 59.38 [c] | 0.22 [bc] | nd | nd | nd | 4.12 [def] |
| PEI | 45.51 [a] | 18.06 [ab] | 60.66 [bc] | 0.22 [bc] | nd | nd | nd | 3.57 [efg] |
| **Day 60** | | | | | | | | |
| **Medium dry matter alfalfa** | | | | | | | | |
| CTR | 36.81 [b] | 18.18 [ab] | 62.36 [ab] | 0.25 [a] | 36.00 [ab] | 41.32 [a] | 9.49 [a] | 4.44 [de] |
| PEI | 36.04 [b] | 18.48 [a] | 65.24 [a] | 0.21 [bc] | 35.24 [b] | 40.70 [b] | 9.67 [a] | 6.37 [c] |
| **High dry matter alfalfa** | | | | | | | | |
| CTR | 44.29 [a] | 18.06 [ab] | 60.72 [bc] | 0.25 [a] | 36.64 [a] | 43.66 [a] | 9.86 [a] | 3.02 [fg] |
| PEI | 44.35 [a] | 18.30 [a] | 58.28 [c] | 0.23 [ab] | 36.24 [ab] | 43.54 [a] | 10.00 [a] | 2.71 [g] |
| SEM | 0.57 | 0.04 | 0.73 | 0.01 | 0.17 | 0.32 | 0.08 | 0.21 |
| **Effects and interactions** | *p*-value | | | | | | | |
| INO | 0.05 | <0.01 | 0.67 | <0.01 | 0.04 | 0.14 | 0.25 | <0.01 |
| DRM | <0.01 | <0.01 | <0.01 | 0.55 | <0.01 | <0.01 | 0.02 | <0.01 |
| DAY | 0.02 | <0.01 | <0.01 | <0.01 | nd | nd | nd | <0.01 |
| INO × DRM | 0.17 | 0.27 | 0.08 | 0.11 | 0.50 | 0.31 | 0.88 | <0.01 |
| INO × DAY | 0.65 | 0.97 | 0.76 | <0.01 | nd | nd | nd | 0.06 |
| DRM × DAY | 0.30 | 0.40 | <0.01 | 0.16 | nd | nd | nd | <0.01 |
| DAY × DRM × INO | 0.36 | 0.08 | <0.01 | 0.01 | nd | nd | nd | 0.02 |

\* CTR—no additive; PEI—inoculated with Purina FI Enhance Inoculant (Purina Animal Nutrition LLC, Arden Hills, MN, USA) (*Pediococcus acidilactici*, *Enterococcus faecium* and *Lactobacillus plantarum*); and Day 3, 30 and 60—days of ensiling. The model included the following main effects: day of ensiling (DAY), effect of inoculation (INO) and effect of DM (DRM). The interactions of DAY × INO, DAY × DRM, INO × DRM and DAY × INO × DRM were tested. SEM—the standard error of mean difference, n = 5. Values in the same column with different following letters (a–g) are significantly different. Means within columns with unlike superscript differ *p* < 0.05; nd—not determined.

**Table 4.** The DM and chemical analysis (% DM basis unless stated otherwise) of alfalfa silage ensiled for 3, 30 and 60 d. Means shown for main effects and two-way interaction when significant. There were no significant INO × DAY or DRM × DAY interactions (data not shown). (ADF, NDF and ash data only for day 60).

| Treatments | Dry Matter | Crude Protein | Acid Detergent Fiber | Neutral Detergent Fiber | Ash |
|---|---|---|---|---|---|
| **INO** | | | | | |
| CTR * | 41.45 | 17.94 [b] | 36.32 [a] | 42.49 | 9.67 |
| PEI | 40.74 | 18.19 [a] | 35.74 [b] | 42.12 | 9.83 |
| **DRM** | | | | | |
| Medium dry matter | 36.96 [b] | 18.25 [a] | 35.62 [b] | 41.01 [b] | 9.58 [b] |
| High dry matter | 45.20 [a] | 17.92 [b] | 36.44 [a] | 43.60 [a] | 9.93 [a] |
| **Day** | | | | | |
| 3 | 41.58 [a] | 17.87 [c] | nd | nd | nd |
| 30 | 41.33 [ab] | 18.07 [b] | nd | nd | nd |
| 60 | 40.37 [b] | 18.26 [a] | 36.03 | 42.30 | 9.76 |
| **INO × DRM** | | | | | |
| Medium dry matter alfalfa | | | | | |
| CTR | 37.57 [b] | 18.03 [b] | 36.00 [ab] | 41.32 [b] | 9.49 [b] |
| PEI | 36.36 [c] | 18.35 [a] | 35.24 [b] | 40.70 [b] | 9.67 [ab] |
| High dry matter alfalfa | | | | | |
| CTR | 45.33 [a] | 17.85 [c] | 36.64 [a] | 43.66 [a] | 9.86 [ab] |
| PEI | 45.12 [a] | 18.03 [b] | 36.24 [b] | 43.54 [a] | 10.00 [a] |
| **Effects and interactions** | | | *p*-value | | |
| INO | 0.05 | <0.01 | 0.04 | 0.14 | 0.25 |
| DRM | <0.01 | <0.01 | <0.01 | <0.01 | 0.02 |
| DAY | 0.02 | <0.01 | nd | nd | nd |
| INO × DRM | 0.17 | 0.27 | 0.50 | 0.31 | 0.88 |

* CTR—no additive; PEI—inoculated with Land O'Lakes inoculant (*Pediococcus acidilactici*, *Enterococcus faecium* and *Lactobacillus plantarum*); and Day 3, 30 and 60—days of ensiling. The model included the following main effects: day of ensiling (DAY), effect of inoculation (INO) and effect of DM (DRM). The interactions of DAY × INO, DAY × DRM, INO × DRM and DAY × INO × DRM were tested. SEM—the standard error of mean difference, n = 5. Values in the same column with different following letters (a–c) are significantly different. Means within columns with unlike superscript differ *p* < 0.05; nd—not determined.

Fermentation end products are shown in Table 5. There was a three-way interaction for pH. Overall, the effect of inoculation on the drop in pH was faster in medium DM silages than high DM silages during the early stages of fermentation. After 3 d of ensiling, medium DM-inoculated silages had markedly lower pH (PEI = 5.17) compared to CTR (5.51), but pH was not affected by inoculation in the high DM treatment. In the current study, after 30 d, inoculated silages were still lower in pH (PEI = 4.08) compared to CTR (4.35) in medium DM silage. After 60 d of ensiling, pH was lower for inoculated silage (average pH = 4.08) vs. CTR (4.29) silage in medium DM. There was a three-way interaction for the concentration of lactic acid in silage (Table 5). Treatment with PEI resulted in consistent, numerically (but not always statistically) higher concentrations of lactic acid when compared to CTR silages at all DM and time points. Specifically, PEI was statistically higher in lactic acid (PEI = 7.04%) than CTR (5.89%) in medium DM silage at 30 d. Treatment with PEI had no effect on the concentrations of acetic acid in high DM silages after 3 d of ensiling (Table 5). However, inoculation resulted in consistently lower concentrations of acetic acid when compared to CTR in medium DM silages after 30 and 60 d. The inoculation of high DM silages numerically, but not statistically, resulted in lower acetic acid than CTR at 30 and 60 d. There was a three-way interaction for 1, 2- propanediol (Table 5). 1, 2- propanediol was present only after 3 d of ensiling. There was only an INO and INO × DAY effect for ethanol, as the PEI treatment (0.87%) was lower than CTR (1.13%) only in medium DM silage (Table 5). When compared to CTR, the numbers of LAB were not greater when

inoculated. This is in contrast to the findings of lower pH in inoculated medium DM silage but suggests that while inoculation did not increase the numbers of LAB, it made fermentation more efficient. Low numbers of LAB in PEI silages in medium DM at 30 d of silage (between 3.6 and 3.94 log cfu/g) are not explainable and appears to be an anomaly. Inoculation had no primary or interaction effects on the numbers of yeasts.

**Table 5.** The pH, fermentation end products (% of DM) and microbial populations (log cfu/g wet weight basis) of alfalfa silage ensiled for 3, 30 and 60 d.

| Treatments | pH | Lactic Acid | Acetic Acid | 1,2-Propane-diol | Propionic Acid | Ethanol | Lactic Acid Bacteria | Yeasts |
|---|---|---|---|---|---|---|---|---|
| **Day 3** | | | | | | | | |
| **Medium dry matter alfalfa** | | | | | | | | |
| CTR * | 5.51 b | 1.61 ef | 0.71 d | 0.00 d | 0.12 bc | 1.12 | 9.12 abc | 3.87 a |
| PEI | 5.17 c | 2.28 e | 0.85 cd | 0.00 d | 0.13 bc | 0.97 | 9.26 ab | 3.77 a |
| **High dry matter alfalfa** | | | | | | | | |
| CTR | 5.69 a | 1.26 f | 0.58 d | 0.00 d | 0.09 bc | 0.87 | 9.72 a | 1.90 cd |
| PEI | 5.73 a | 1.24 f | 0.60 b | 0.00 d | 0.10 bc | 0.92 | 9.22 abc | 1.69 d |
| **Day 30** | | | | | | | | |
| **Medium dry matter alfalfa** | | | | | | | | |
| CTR | 4.35 f | 5.89 c | 2.40 ab | 0.18 c | 0.35 a | 1.12 | 7.12 cd | 3.02 abc |
| PEI | 4.08 h | 7.04 ab | 1.24 c | 0.00 d | 0.15 bc | 0.80 | 3.60 e | 3.00 abc |
| **High dry matter alfalfa** | | | | | | | | |
| CTR | 4.70 d | 4.04 d | 2.37 ab | 0.39 b | 0.18 bc | 1.04 | 8.80 abc | 2.91 abc |
| PEI | 4.64 d | 3.75 d | 2.07 b | 0.36 b | 0.19 b | 0.90 | 8.94 abc | 2.59 bcd |
| **Day 60** | | | | | | | | |
| **Medium dry matter alfalfa** | | | | | | | | |
| CTR | 4.29 g | 6.44 bc | 2.57 a | 0.28 bc | 0.05 c | 1.15 | 7.50 bcd | 2.75 abcd |
| PEI | 4.08 h | 7.32 a | 2.06 b | 0.17 c | 0.34 a | 0.83 | 5.58 de | 3.08 ab |
| **High dry matter alfalfa** | | | | | | | | |
| CTR | 4.53 e | 4.13 d | 2.57 a | 0.66 a | 0.07 bc | 0.96 | 8.51 abc | 3.30 ab |
| PEI | 4.50 e | 4.17 d | 2.50 ab | 0.63 a | 0.18 b | 0.94 | 8.67 abc | 2.99 abc |
| SEM | 0.07 | 0.28 | 0.11 | 0.03 | 0.01 | 0.03 | 0.25 | 0.10 |
| **Effects and interactions** | | | | | *p*-value | | | |
| INO | <0.01 | <0.01 | <0.01 | <0.01 | 0.01 | <0.01 | <0.01 | 0.54 |
| DRM | <0.01 | <0.01 | 0.01 | <0.01 | <0.01 | 0.21 | <0.01 | 0.39 |
| DAY | <0.01 | <0.01 | <0.01 | <0.01 | <0.01 | 0.99 | <0.01 | <0.01 |
| INO × DRM | <0.01 | <0.01 | <0.01 | 0.02 | 0.64 | 0.02 | 0.02 | 0.03 |
| INO × DAY | 0.08 | 0.85 | <0.01 | 0.03 | <0.01 | 0.31 | 0.06 | 0.14 |
| DRM × DAY | <0.01 | <0.01 | <0.01 | <0.01 | 0.50 | 0.37 | <0.01 | <0.01 |
| DAY × DRM × INO | <0.01 | 0.29 | <0.01 | 0.15 | <0.01 | 0.84 | <0.01 | <0.01 |

* CTR—no additive; PEI—inoculated with Land O'Lakes inoculant (*Pediococcus acidilactici*, *Enterococcus faecium* and *Lactobacillus plantarum*); and Day 3, 30 and 60—days of ensiling. The model included the following main effects: day of ensiling (DAY), effect of inoculation (INO) and effect of DM (DRM). The interactions of DAY × INO, DAY × DRM, INO × DRM and DAY × INO × DRM were tested. SEM—the standard error of mean difference, n = 5. Values in the same column with different following letters (a–h) are significantly different. Means within columns with unlike superscript differ *p* < 0.05.

*3.4. Analysis of the Composition of the Bacterial and Fungal Communities by Next-Generation Sequencing for Fresh Alfalfa and 90 d Silage*

The alpha diversity of bacterial and fungal communities for alfalfa silage is shown in Table S1. Fresh forage had a similar diverse bacterial population with silage, as indicated by a more or less consistent Shannon index with silages (Supplementary Table S1). Regarding the fungal community diversity, silages had a low Shannon and Chao1 index than fresh forage, indicating that ensiling reduced fungal diversity. For a better understanding of the microbial community structure in alfalfa silage, the 10 most abundant family and genus of bacteria and fungi in fresh forage and silages can be found in Figures 1 and 2.

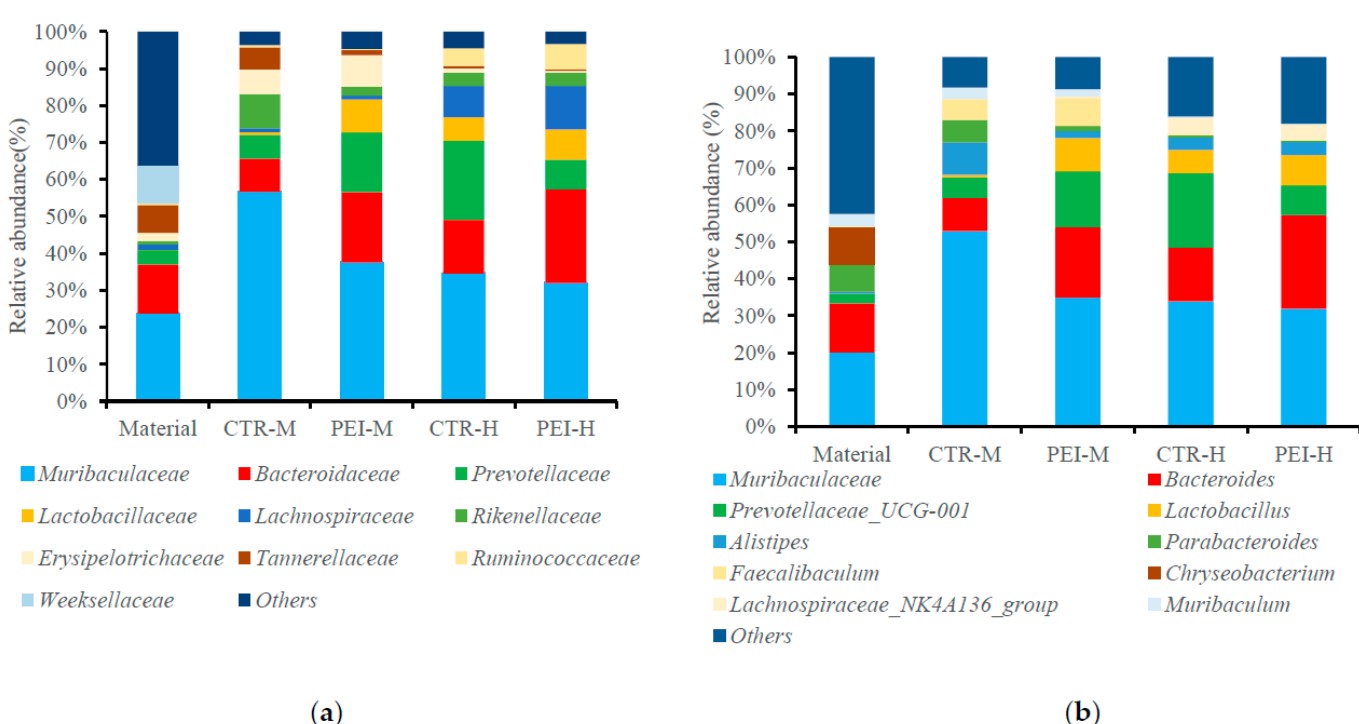

**Figure 1.** Relative abundances of the 10 most abundant family and genus of the bacterial microbiota of pre-ensiled crops and silages prepared from alfalfa silage. (**a**) Family- and (**b**) genus-level bacterial community in alfalfa silage. M—alfalfa ensiled at medium DM; H—alfalfa ensiled at high DM; CTR—no additive; PEI—inoculated with Land O'Lakes inoculant (*Pediococcus acidilactici*, *Enterococcus faecium* and *Lactobacillus plantarum*); M—medium DM alfalfa silage; and H—high DM alfalfa silage.

The predominant bacterial genera were *Muribaculaceae* and *Bacteroides* in fresh forage and silages, which were members of the order *Bacteroidales* (Figure 1). While *Lactobacillus* (6.04%) was the fourth predominant bacterial taxa in silages, traces of *Lactobacillus* were detected in material. At the genus level, the abundance of *Muribaculaceae* (20%) and *Bacteroides* (13%) was higher than other bacteria in fresh material. Sequences from *Chryseobacterium*, *Parabacteroides* and *Muribaculum* accounted for 9%, 7% and 4% abundance in fresh material, respectively. After 60 d ensiling, at the genus level, in medium dry matter silages, the most abundant bacterial populations were *Muribaculaceae*, *Bacteroides*, *Alistipe*, *Parabacteroides*, *Prevotellaceae* and *Faecalibaculum* accounting for 53%, 9%, 9%, 6%, 6% and 6% of all the bacterial reads, respectively. Moreover, the abundance of *Lactobacillus* (0.7%) was detected in medium DM CTR silage. For medium DM PEI silage, *Muribaculaceae* (35%), *Bacteroides* (19%) and *Prevotellaceae* (15%) were the three most abundant genera for the bacterial community, which accounted for over 70% of total sequences (Figure 1b). The percentage of *Lactobacillus* (9% for PEI) was distinctly higher than those of medium DM CTR silage. For the high dry matter alfalfa silage bacterial community, at the genus level, *Muribaculaceae* (34%), *Bacteroides* (15%), *Prevotellaceae* (20%) and *Lactobacillus* (6%) were the four most abundant genera in high DM CTR silage. *Muribaculaceae* (32%), *Bacteroides*

(25%), *Prevotellaceae* (8%) and *Lactobacillus* (8%) were the most predominant bacterial taxa in PEI silages.

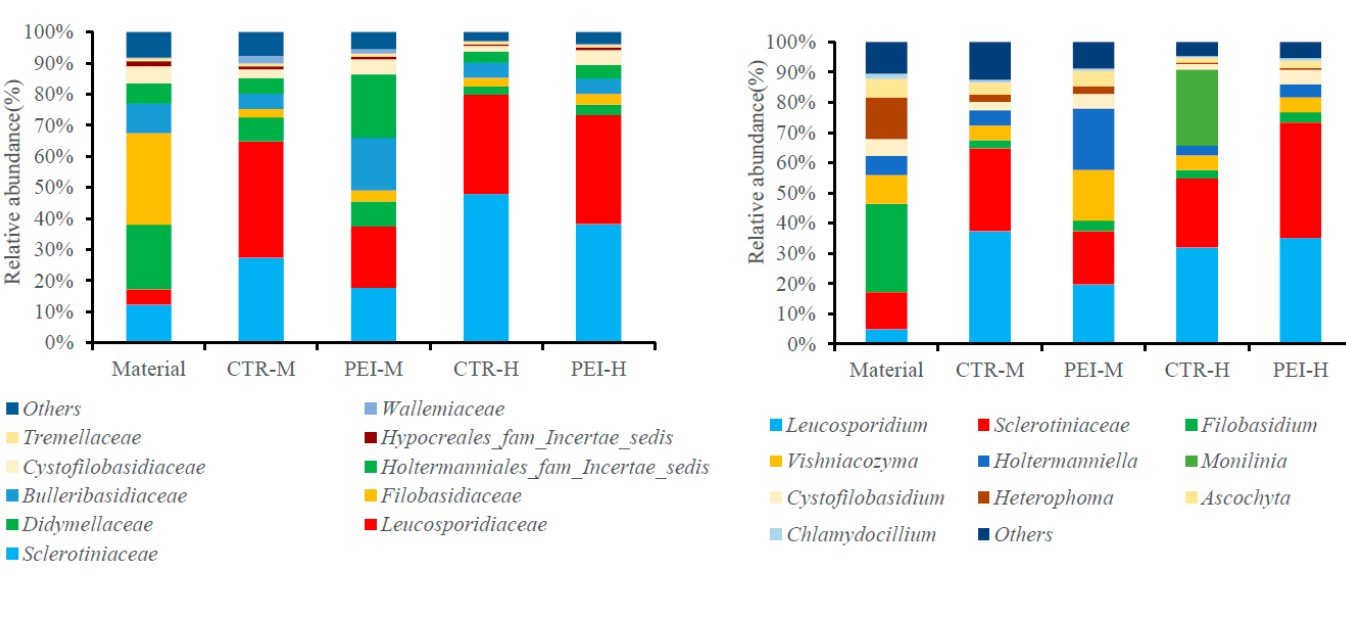

**Figure 2.** Relative abundances of the 10 most abundant family and genus of the fungal microbiota of pre-ensiled crops and silages prepared from alfalfa silage. (**a**) Family- and (**b**) genus-level bacterial community in alfalfa silage. M—alfalfa ensiled at medium DM; H—alfalfa ensiled at high DM; CTR—no additive; PEI—inoculated with Purina FI Enhance Inoculant (PEI, Purina Animal Nutrition LLC, Arden Hills, MN, USA) (*Pediococcus acidilactici*, *Enterococcus faecium* and *Lactobacillus plantarum*); M—medium DM alfalfa silage; and H—high DM alfalfa silage.

Figure 2 also shows the 10 most abundant fungal genera in fresh forage and silages. Silages have different predominant fungal genera compared with fresh material. *Leucosporidium* and *Sclerotiniaceae* were the most primary fungi in silages, while *Filobasidium* and *Heterophoma* dominated the alfalfa material. After 90 d ensiling, at the genus level, in medium DM CTR silages, the most predominant fungal populations were found to be *Leucosporidium* (37%) and *Sclerotiniaceae* (27%), which accounted for over 64% of total sequences (Figure 2b). For medium DM PEI silages, the most abundant fungal populations were *Leucosporidium*, *Sclerotiniaceae* and *Vishniacozyma* accounted for 20%, 18% and 17%, respectively. For high DM CTR silages, *Leucosporidium* (32%) was the most predominant fungal taxa, followed by *Sclerotiniaceae* (23%). For high DM PEI silages, *Sclerotiniaceae* was a dominant genus detected with 38% abundance, followed by *Leucosporidium* (35%) (Figure 2b), while other genera were detected below 5% of their reads.

Table 6 shows the effect of dry matter content and inoculation on the four most abundant bacterial and fungal genera in the alfalfa silages. For bacterial genera, the additive significantly affected the RA of *Bacteroides* and *Lactobacillus* ($p < 0.05$). The dry matter difference affected the RA of *Bacteroides* and *Lactobacillus* ($p < 0.05$) in silages. There was an interaction between dry matter content and additive for the RA of *Muribaculaceae*, *Bacteroides*, *Prevotellaceae* and *Lactobacillus* ($p < 0.05$) in silages. In contrast, the additive caused a marked decrease in the RA of *Prevotellaceae* in high DM PEI compared to high DM CTR. The addition of bacterial additives did not affect the numbers of cultivable LAB on MRS when compared to CTR in PEI silage, but it increased ($p < 0.05$) the RA of *Lactobacillaceae* in medium DM PEI compared to CTR. Table 6 shows the effect of dry matter content and additive treatment on the four most abundant fungal genera in the alfalfa silages. The addition of LAB and dry matter values affect the RA of *Vishniacozyma*

($p < 0.01$). There was an interaction between the dry matter content and additive for the RA of *Leucosporidium*, *Sclerotiniaceae* ($p < 0.05$) and *Vishniacozyma* ($p < 0.01$) in silages.

**Table 6.** Relative abundance (%) of material, bacterial genus and fungal genus in alfalfa silages ensiled for 60 d, as analyzed by the sequencing of the V4–V5 region of the 16S rRNA, for bacteria, and ITS1, for fungi, using the Illumina MiSeq platform.

| Item | \multicolumn{4}{c}{**Bacterial Genus**} | | | | \multicolumn{4}{c}{**Fungal Genus**} | | | |
|---|---|---|---|---|---|---|---|---|
| | Muribaculaceae | Bacteroides | Prevotellaceae-UCG-001 | Lactobacillus | Leucosporidium | Sclerotiniaceae | Filobasidium | Vishniacozyma |
| Material | 20.08 | 13.27 | 2.61 | 0.05 | 4.91 | 12.28 | 29.32 | 9.43 |
| **Medium dry matter alfalfa** | | | | | | | | |
| CTR * | 52.86 [a] | 8.96 [c] | 5.66 [b] | 0.72 [b] | 37.40 [a] | 27.40 [ab] | 2.68 | 4.89 [b] |
| PEI | 34.94 [b] | 19.04 [ab] | 15.13 [a] | 8.95 [a] | 19.72 [b] | 17.72 [b] | 3.45 | 16.74 [a] |
| **High dry matter alfalfa** | | | | | | | | |
| CTR | 33.88 [b] | 14.54 [b] | 20.15 [a] | 6.32 [a] | 32.00 [a] | 22.86 [b] | 2.75 | 4.81 [b] |
| PEI | 31.87 [b] | 25.38 [a] | 8.03 [b] | 8.18 [a] | 35.16 [a] | 38.19 [a] | 3.44 | 4.86 [b] |
| SEM | 4.20 | 2.64 | 2.14 | 1.52 | 3.26 | 3.61 | 2.46 | 1.07 |
| **Effects and interactions** | \multicolumn{8}{c}{*p*-value} | | | | | | | |
| INO | 0.30 | 0.03 | 0.78 | 0.04 | 0.24 | 0.58 | 0.26 | <0.01 |
| DRM | 0.25 | 0.04 | 0.45 | 0.04 | 0.41 | 0.13 | 0.97 | <0.01 |
| INO × DRM | 0.04 | 0.04 | 0.04 | 0.03 | 0.04 | 0.03 | 0.95 | <0.01 |

* CTR—no additive; PEI—inoculated with Land O'Lakes inoculant (*Pediococcus acidilactici*, *Enterococcus faecium* and *Lactobacillus plantarum*); and Day 3, 30 and 60—days of ensiling. The model included the following main effects: effect of inoculation (INO) and effect of DM (DRM). The interactions of INO × DRM were tested. SEM—the standard error of mean difference, n = 5 or 4. Values in the same column with different following letters (a–c) are significantly different. Means within columns with unlike superscript differ $p < 0.05$.

## 4. Discussion

The chemical and microbiological content of freshly chopped forages are shown in Table 2. The nutrient components were considered "normal" for alfalfa and, with the exception of DM, were very similar between the two DMs. The differences in alfalfa DM between treatments before ensiling were biologically significant enough to affect the silage characteristics ($p < 0.05$). Low numbers of *lactobacilli* and high numbers of aerobic bacteria were present in alfalfa material. This was observed previously by Cai et al. (2017) [29], who suggested that this may result in poor fermentation. Therefore, it is necessary to add fermentable substrates and LAB inoculants to control microbes in silage fermentation [30].

The benefits of LAB at ensiling were shown again, in this study, where the application of homo-LAB improved silage fermentation by lowering silage pH and the concentration of ammonia-N and increased WSC concentration and a higher concentration of lactic acid in medium DM silages. Lower CP and higher NDF in high DM silages is not surprising as there is greater potential for leaf loss during the harvesting of high DM alfalfa. Generally, the higher production of lactic acid via inoculation supports the findings that lower pH with inoculation is a strong indicator that the added lactic acid bacteria dominated the overall fermentation. These results indicated that application of homo-LAB resulted in silage with a more homo-lactic profile in medium DM silages [31]. A faster drop in silage pH is desirable because low pH can inhibit undesirable *Enterobacteria* and *Clostridia* [32]. The lack of consistent improvements in fermentation in high DM silages treated with homo-LAB is not unexpected because fermentation becomes more restricted as water activity declines with increasing concentrations of DM in silages [23]. Previous work in our lab supports that, in high DM alfalfa silages, there is a lag of several days prior to active fermentation, most likely because of the low available water for microbial growth [30]. Treatment with homo-LAB did not result in the classical effects of more lactic and less acetic acid and the suppression of proteolysis often reported in previous studies [10,30].

At the genus level, *Muribacula* (20.08–52.86%), *Bacteroides* (8.96–25.38%), *Prevotellaceae* (2.61–20.15%) and *Lactobacillus* (0.05–8.95%) were dominant in all samples. Other genera

(*Alistipes*, *Parabacteroides*, *Faecalibaculum*, *Chryseobacterium*, *Lachnospiraceae* and *Muribaculum*) were represented with lower abundance but occupied the top 10 genera in material and silages (Figure 1b). Several predominant bacterial genera of epiphytic flora survived and were represented with a higher abundance. Inoculants comprising certain strains of LAB have been developed to reduce the reliance of epiphytic flora, but *Lactobacillus* was not the most abundant genus in all silages. Interestingly, in this study, the structure of the silage flora was similar with the flora in the rumen. To our knowledge, this is the first study to find that the abundance of rumen flora (like top 10 genus) was generally significantly higher in silages than in fresh material ($p < 0.05$). This indicated that these rumen bacterial genera can survive, reproduce and be the predominant bacterial genera in silages. In previous studies, the additives used generally have not had consistent effects on nutrient composition [33,34]. In the current study, the NDF and ADF concentration of silages was altered by the homo-LAB treatment and dry matter change. This finding might have been a result of the main effect of these rumen bacterial genera in silages. It is known that *Prevotellaceae*, *Bacteroides* and *Muribaculaceae*, which belong to the phylum *Bacteriodetes*, responsible for hemicellulose, pectin and high carbohydrate levels, feed digestion and are capable of degrading cellulose and starch in the rumen [35,36]. *Bacteroides* is an important anaerobic genus that plays a fundamental role in the gut ecosystem by breaking down polysaccharides [37]. It consumes lactic acid and converts it to VFA and is able to degrade deoxy sugars via the propanediol pathway but produce the pathway intermediate 1, 2-propanediol as the final product [38]. The appearance of 1, 2-propanediol and propionic acid in silages may be attributed to these rumen bacteria. The increased population of the genus *Prevotellaceae* in silages produced with LAB inoculants is another explanation for the higher molar proportion of propionic acid in medium DM PEI silages.

The homo-LAB inoculant was successful in lowering pH and increasing lactic acid concentrations in silages but did not affect the yeast population and RA of fungi. The DM content also had no effect on the yeast exception of *Vishniacozyma*. *Leucosporidium* sp. was the first, most abundant feature in all the silage samples (Table 6). *Leucosporidium* is able to assimilate glucose [39]. *Sclerotiniaceae* usually is found in alfalfa and causes disease because of its ability for long-term survival accompanied by great reproduction potential [40]. *Filobasidium* is relatively common among the species of endophytic fungi of grasses and has been isolated from the rhizosphere of corn [41]. The genus *Filobasidium* has been reported to be an abundant genus in fresh corn silage and declines with ensiling time [42]. *Vishniacozyma* (also known as *Cryptococcus*) has been isolated from soil and wheat [43] while there are no reports of this genus in silage. *Vishniacozyma* was reported to be able to assimilate lactic acid and D-lactose [44]. Most yeast comes from soil. Ash is a measure of the forage, with values > 10% for grasses reflecting soil contamination [45]. Higher levels of ash in silages could be attributed to an increased risk of soil contamination when grasses are transported from the field to the farm in this study.

## 5. Conclusions

Silage is heterogenous in nature with differences in chemical and microbiological composition, as well as chemical properties within a silo [46], thus, different kinds of silage types are worth exploring in depth. The results from this study show that using a homo-LAB inoculant resulted in a lower drop in pH especially when applied to medium DM alfalfa silage and, in general, resulted in a more efficient silage fermentation characterized by higher lactic acid and lower acetic acid. Inoculating medium DM alfalfa also resulted in lower $NH_3$-N and higher residual WSC concentrations compared to CTR after 60 d of fermentation. Overall, treatment with the inoculant in this study was highly beneficial as it caused a faster drop in pH and a more homo-lactic acid type of fermentation. The changes in the relative abundance of the main populations of bacteria and yeasts did explain the improvements in fermentation products and nutrition changes by the additive application. The finding that the rumen flora may survive and dominate in alfalfa silages needs to be further studied. The assessment and further study of homo- or hetero-LAB inoculants for

other fodder plants is also necessary to improve our knowledge and understanding about the role of microorganisms in the ensiling process.

**Supplementary Materials:** The following supporting information can be downloaded at: https://www.mdpi.com/article/10.3390/fermentation8110621/s1, Table S1: alpha diversity of bacterial and fungal community for alfalfa silage.

**Author Contributions:** L.K.J. conceived and designed the research, contributed to manuscript revision and read and approved the submitted version. Y.L. and E.B.d.S. conducted experiments and analyzed the data. J.L. contributed analytical tools. Y.L. and L.K.J. wrote the first draft of the manuscript. All authors have read and agreed to the published version of the manuscript.

**Funding:** This study was supported by the Joint Fund for National Natural Science Foundation of China (Grant No. 31402136).

**Institutional Review Board Statement:** Not applicable.

**Informed Consent Statement:** Not applicable.

**Data Availability Statement:** The data presented in this study are available on request from the first author. The data are not publicly available due to restrictions by the research group. The data presented in the study are deposited in the Sequence Read Archive (SRA), accession number PRJNA886387. We have released the above data.

**Acknowledgments:** This research was partially supported by Purina Animal Nutrition LLC, Shoreview, MN. The authors thank the farm staff of the University of Delaware farm for assistance with forage harvesting and Michelle Der Bedrosian for silo preparation.

**Conflicts of Interest:** The authors declare no conflict of interest.

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
