# Peer review of "Effect of Homo-Fermentative Lactic Acid Bacteria Inoculants on Fermentation Characteristics and Bacterial and Fungal Communities in Alfalfa Silage"

_fermentation, doi:10.3390/fermentation8110621_

Round 1

Reviewer 1 Report

Specific reasons for selecting these two different DM values of alfalfa have to be inserted in the introduction. Why you selected alfalfa silage, as corn silage is the most common type used widely? The main issue for me is how did you adjust the DM at two different degrees? If you used the same alfalfa variety in both DM degrees, it means you collected the plants at two diffrent life cycles, if so the chemical composition of the alfalfa before treatments can not be the same, I wonder the data in table 1, if they were different variety thus many other factors can affect the produced silage quality as carbohydrates (either fibers or soluble ones) and it will interact with the action of the treatment.

In the title of the manuscript, please don't add abbreviations, write the full name of any abbreviations throughout the manuscript in their first mentions.

Line 10 seems to be unclear, I suggest the following modification: alfalfa silage ensiled at two different dry matter (DM) content (38 and 46% DM). Both alfalfa types were treated or not with.......

Line 15, please don't start the sentence with an abbreviation.

Line 18, what do you mean by CTR-M and  CTR-H, please insert the full names of all abbreviations in their first mentions.

Line 48, It seems some words are missing here.

Line 60, more information is needed about the experimental alfalfa, it seems that they are different in their life cycle, did you use the same variety? Please insert more information about the lactic acid (e.g., source, preparation, media, concentration,....).

for Statistical Analysis, please add the number of statistical repetitions for each group.

Please try to reduce the abbreviations in the tables for more clarity..

I don't understand why you used aNDFom and NDF?  more important to include the ADL, EE, and NFE, and keep only the aNDFom!! then in this case calculate the ADF also to be aADFom.

Values in all tables have to be in three digits, the significance letters have to be superscripted. 

Author Response

Point 1: Specific reasons for selecting these two different DM values of alfalfa have to be inserted in the introduction. Why you selected alfalfa silage, as corn silage is the most common type used widely? The main issue for me is how did you adjust the DM at two different degrees? If you used the same alfalfa variety in both DM degrees, it means you collected the plants at two diffrent life cycles, if so the chemical composition of the alfalfa before treatments can not be the same, I wonder the data in table 1, if they were different variety thus many other factors can affect the produced silage quality as carbohydrates (either fibers or soluble ones) and it will interact with the action of the treatment.

Response 1: As Reviewer 1 suggested, we have checked the Introduction, and rewrote the introduction used publications on the broader spectrum of LABs and silage preservatives. We also explained why we selected alfalfa with two DM values “Alfalfa is one of the popular fodder crops fed to ruminants, while good quality alfalfa silage is difficult to make due to the high buffering capacity, low concentration of soluble carbohydrates and the presence of epiphytic lactic acid bacteria in the raw material [1].”“ For example in alfalfa silages wetter than 40% DM, Stallings et al. (1981) and Papadopoulos and McKersie (1983) reported increased proteolysis than in more highly wilted silages, while, in drier silages (greater than 40% DM), increased wilting has consistently reduced proteolysis [6,7].”  (L35-76).

We judged the DM of alfalfa during wilting by Microwaves which we measured DM of wilting alfalfa once an hour.

We used alfalfa at same life cycles, DM difference by wilting. We rewrote the method “Second-cut alfalfa (early bloom stage) from one field was harvested at the University of Delaware farm in Newark, Delaware and wilted to a DM content of 37.8% or 46.5%.”(L80-81).

Point 2: In the title of the manuscript, please don't add abbreviations, write the full name of any abbreviations throughout the manuscript in their first mentions.

Response 2: We have modified the text. (L4).

Point 3: Line 10 seems to be unclear, I suggest the following modification: alfalfa silage ensiled at two different dry matter (DM) content (38 and 46% DM). Both alfalfa types were treated or not with.......

Response 3: We have modified the text. (L14-16).

Point 4: Line 15, please don't start the sentence with an abbreviation.

Response 4: We have modified the text. (L19-20).

Point 5: Line 18, what do you mean by CTR-M and CTR-H, please insert the full names of all abbreviations in their first mentions.

Response 5: We have moved the sentence and modified the text. (L20-27).

Point 6: Line 48, It seems some words are missing here.

Response 6: We have modified the text. (L35-76).

Point 7: Line 60, more information is needed about the experimental alfalfa, it seems that they are different in their life cycle, did you use the same variety? Please insert more information about the lactic acid inoculant (e.g., source, preparation, media, concentration,....).

Response 7: We used alfalfa at same life cycles, DM difference by wilting. We rewrote the method “Second-cut alfalfa (early bloom stage) from one field was harvested at the University of Delaware farm in Newark, Delaware and wilted to a DM content of 37.8% or 46.5%.”(L79-81).

Lactic acid inoculant: Purina FI Enhance Inoculant (PEI, Purina Animal Nutrition LLC, Arden Hills, Minnesota, USA) containing Pediococcus acidilactici, Enterococcus faecium and Lactobacillus plantarum (L83-85).

Point 8: for Statistical Analysis, please add the number of statistical repetitions for each group.

Response 8: We have modified the text. (L173-174).

Point 9: Please try to reduce the abbreviations in the tables for more clarity..

Response 9: We have modified the text. (L298-351).

Point 10: I don't understand why you used aNDFom and NDF?  more important to include the ADL, EE, and NFE, and keep only the aNDFom!! then in this case calculate the ADF also to be aADFom.

Response 10: We have moved the aNDFom and modified the text (L298-351).

Point 11: Values in all tables have to be in three digits, the significance letters have to be superscripted. 

Response 11: We have modified the text (L298-351).

Reviewer 2 Report

Dear Editor and Authors,

the manuscript titled "Effect of Homo-LAB Inoculants on Fermentation Characteristics, VOCs and Bacterial and Fungal Communities in Alfalfa Silage" is very good paper and complements the knowledge on strategic forage conservation. The work requires a few minor corrections to be accepted for publication.

Abstract: The abstract is very well written, but I suggest that in the text - numbers 1 to 9 (excluding: sequences of numbers, numbers in the name and recipes) should be written in words (abstract line 11, line 304 and other palces).

It is not clear from the abstract what CTR-M and CTR-H are, but it is necessary. I know you have reached the abstract word limit, however you could only describe the best option .

On the other hand, you can explain these abbreviations at the beginning of the abstract, which will limit further problems.

Introduction: The introduction is very poor. You have posted some papers on legumes, but you definitely need to rely on publications on the broader spectrum of LABs and silage preservatives.

I suggest: DOI: 10.1002 / jsfa.10999 (The fact that biostimulants applied already at the stage of cultivation can support the fermentation of green fodder).

DOI: doi.org/10.3390/fermentation8060285 (Very good work, referring to your research, also maize, so you can compare alfalfa with model silage).

DOI: 10.1002 / jsfa.10126 (The fact that various additives of different LABs have different effects on the quality of silage and complex LAB compositions [homo / hetero] work well, especially in reducing undesirable microbiota).

Materials and methods: The entire chemical analysis complies with the requirements for the characteristics of the quality of the silage. I'm not sure bold shortcuts are a good idea.

Section 2.3. Lines 115, 124. Explain why did you choose such temperatures? The optimal temperature for molds is 24-28 degrees Celsius. Environmental LABs prefer 30 degrees Celsius (higher temperatures are for LABs inhabiting mammals).

Section 2.4 Lines 126-131. Just enough? How was the DNA isolation procedure? Have you used PMA dye to analyze the DNA of only living bacteria? What platform did you perform the sequencing on. Basic data must be written!

Section 2.5. You used Tukey's test, but did you confirm that the parametric statistic was selected correctly: that is, the distribution was normal and did you confirm the homogeneity of the variance (Levene test) ??

Results: Section 3.4. L 222: You write that you presented the alpha-diversity in table A1, but I do not see the description of these results here, because after this sentence you go to the description of domination, not indicators.

 Discussion and conclusions: The discussion is generally good, while in the conclusions you have to further emphasize the good points of homo-LAB for alfalfa; maybe also in comparison with other fodder plants and other solutions (preservatives).

Figures: It would be good to correct the figures - they are artificially stretched which reduces their readability.

Tables: Tab 3. It is necessary to explain what the four dashes mean in some parts of the table, everything in the tables must be clear.

Author Response

Point 1: the manuscript titled "Effect of Homo-LAB Inoculants on Fermentation Characteristics, VOCs and Bacterial and Fungal Communities in Alfalfa Silage" is very good paper and complements the knowledge on strategic forage conservation. The work requires a few minor corrections to be accepted for publication.

Response 1: We have modified the text.

Point 2: Abstract: The abstract is very well written, but I suggest that in the text - numbers 1 to 9 (excluding: sequences of numbers, numbers in the name and recipes) should be written in words (abstract line 11, line 304 and other palces).

Response 2: We have modified the text (L14-27;32-76).

Point 3: It is not clear from the abstract what CTR-M and CTR-H are, but it is necessary. I know you have reached the abstract word limit, however you could only describe the best option .

Response 3: We have modified the text. (L14-27).

Point 4: On the other hand, you can explain these abbreviations at the beginning of the abstract, which will limit further problems.

Response 4: We have modified the text (L14-27).

Point 5: Introduction: The introduction is very poor. You have posted some papers on legumes, but you definitely need to rely on publications on the broader spectrum of LABs and silage preservatives.

I suggest: DOI: 10.1002 / jsfa.10999 (The fact that biostimulants applied already at the stage of cultivation can support the fermentation of green fodder).

DOI: doi.org/10.3390/fermentation8060285 (Very good work, referring to your research, also maize, so you can compare alfalfa with model silage).

DOI: 10.1002 / jsfa.10126 (The fact that various additives of different LABs have different effects on the quality of silage and complex LAB compositions [homo / hetero] work well, especially in reducing undesirable microbiota).

Response 5: As Reviewer 1 suggested, we had better have checked the Introduction, and rewrote the introduction used publications on the broader spectrum of LABs and silage preservatives. (L32-76).

Point 6: Materials and methods: The entire chemical analysis complies with the requirements for the characteristics of the quality of the silage. I'm not sure bold shortcuts are a good idea.

Response 6: We have modified the text (L32-76).

Point 7: Section 2.3. Lines 115, 124. Explain why did you choose such temperatures? The optimal temperature for molds is 24-28 degrees Celsius. Environmental LABs prefer 30 degrees Celsius (higher temperatures are for LABs inhabiting mammals).

Response 7: We usually incubate at 30C so this should be corrected.  Also 25 to 30C is the recommended incubation temperature from Oxoid.

Point 8: Section 2.4 Lines 126-131. Just enough? How was the DNA isolation procedure? Have you used PMA dye to analyze the DNA of only living bacteria? What platform did you perform the sequencing on. Basic data must be written!

Response 8: We have modified the text (L144-165).

Point 9: Section 2.5. You used Tukey's test, but did you confirm that the parametric statistic was selected correctly: that is, the distribution was normal and did you confirm the homogeneity of the variance (Levene test) ?

Response 9: We arcsine-transformed all the observations before statistical analysis and

 tested the normally distributions and homogeneity of variance of the data by using Kolmogorov– Smirnov test and Levene test respectively, which were the assumptions

 of Tukey’s test. All P-values were greater than 0.05 which proved that we could use 

Tukey’s test in Section 2.5. Thanks for your suggestions. I will describe the statistical methods and the corresponding results in more detail. (L166-179).

Point 10: Results: Section 3.4. L 222: You write that you presented the alpha-diversity in table A1, but I do not see the description of these results here, because after this sentence you go to the description of domination, not indicators.

Response 10: We have modified the text (L245-251).

Point 11: Discussion and conclusions: The discussion is generally good, while in the conclusions you have to further emphasize the good points of homo-LAB for alfalfa; maybe also in comparison with other fodder plants and other solutions (preservatives).

Response 11: We have modified the text (L426-440).

Point 12: Figures: It would be good to correct the figures - they are artificially stretched which reduces their readability.

Response 12: We have modified the text (L302-308).

Point 13: Tables: Tab 3. It is necessary to explain what the four dashes mean in some parts of the table, everything in the tables must be clear.

Response 13: We have modified the text (L298-351).

Round 2

Reviewer 1 Report

However, I got some difficulties following the corrections because the authors did not write them in different colors (e.g., red) but the manuscript is now improved and the authors followed most of my suggestions.

Only the values in all tables have to be justified in three digits, some tables are three, and others 4, please unify them.

Regards,